# Effects of Non-Native Annual Plant Removal on Native Species in Mediterranean-Climate Shrub Communities

Priscilla M. Ta [1], Emily Griffoul [2], Quinn Sorenson [3], Katharina T. Schmidt [1], Isaac Ostmann [3], Travis E. Huxman [1,2], Jennifer J. Long [1], Kathleen R. Balazs [1,3], Jutta C. Burger [3], Megan Lulow [3] and Sarah Kimball [1,*]

1 Center for Environmental Biology, University of California, Irvine, CA 92697-2525, USA; pmta@uci.edu (P.M.T.); ktschmid@uci.edu (K.T.S.); thuxman@uci.edu (T.E.H.); jjlong@uci.edu (J.J.L.); kathleen.balazs@gmail.com (K.R.B.)
2 Department of Ecology & Evolutionary Biology, University of California, Irvine, CA 92697-2525, USA; emily.griffoul@gmail.com
3 Irvine Ranch Conservancy, 4727 Portola Parkway, Irvine, CA 92620-1914, USA; qmsorenson@gmail.com (Q.S.); iostmann@irconservancy.org (I.O.); jburger@cal-ipc.org (J.C.B.); mlulow@uci.edu (M.L.)
* Correspondence: skimball@uci.edu

**Abstract:** Removal of non-native plants is known to increase overall native cover within degraded communities that contain at least a small percentage of native plant cover. We investigated the mechanisms behind this pattern, asking whether removal of non-native annual species increases the density and species richness of the native community through increased seedling recruitment or through the growth of established native shrubs. We also investigated whether the effectiveness of non-native removal was influenced by region (coastal versus inland) and whether there was a threshold of native cover required for invasive removal to be effective. We established 13 study sites (7 coastal and 6 inland) located throughout the Nature Reserve of Orange County, CA, USA. Each degraded site contained four paired plots corresponding to a range of existing native plant cover: low 20–29%, medium-low 30–39%, medium-high 40–49%, and high cover 50–59% with one plot per pair subjected to non-native removal. We collected plant density, species richness, and established native shrub volume measurements to clarify the effectiveness of non-native removal. Non-native plant removal reduced non-native annual recruitment, increased that of native shrub seedlings, but had no impact on native forb recruitment. Non-native removal increased the number and reduced mortality of established native shrubs but did not influence shrub size. Native seedling density, species richness, and established native shrub number were highest inland, but coastal sites had larger adult shrubs. We found that non-native removal was most effective for increasing native density and species richness for degraded inland sites with less than 40% of existing native cover. The initial native cover did not affect established shrub volume or number. Our results confirm the importance of non-native plant removal in areas with medium-low or low native cover to increase native recruitment, species richness, adult shrub number, and to reduce established shrub mortality, especially during extreme drought.

**Keywords:** coastal sage scrub; natural regeneration; non-native removal; seedling density; species richness; shrub growth; threshold

## 1. Introduction

The Mediterranean biome, which supports high levels of endemism and plant biodiversity, has experienced great loss of biodiversity primarily due to climate and land use change [1–3]. Anthropogenic impacts, such as land development, increased fire frequency, nitrogen deposition, climate-change induced drought, and over a century of intensive grazing, threaten plant communities around the globe and have resulted in an increase in the cover of non-native species. In our coastal sage scrub system, this has led to a conversion of native shrublands into non-native annual grasslands [4–6]. Healthy coastal sage

scrub primarily consists of drought-deciduous shrubs with a diverse native herbaceous understory, yet most coastal sage scrub communities have been invaded by non-native annual grasses and forbs, leading to a hybrid non-native grass–native shrub community. Non-native annuals are not necessarily harmful to an undisturbed, well-established shrub community, but when combined with disturbances, such as prologued drought and increasingly frequent wildfires, they can be highly invasive, eventually leading to vegetation-type conversion from a perennial native shrub-dominated community to an invasive annual community [5,6]. Land stewards have tested multiple ecological restoration methods to reverse the effects of degradation and return annual non-native grass-dominated landscapes to native shrubs [7,8].

Restoration efforts often focus on removal of the non-native annuals, either by hand weeding or herbicide application. When vegetation type-conversion from native shrubland to a non-native annual community has already occurred, such as in the case of heavily degraded communities where key ecosystem processes are altered, more robust active techniques involving the addition of native plants, seeds, or even soil is necessary [9–11]. At lower levels of non-native cover, if important ecosystem processes have not yet been impacted and there is a remnant intact native plant community, then non-native removal alone would be just as effective and even preferred [4,7,11–13]. Non-natives may not be a threat to the native community without additional disturbance, and removal efforts can result in the accidental removal of natives [14]. At some threshold level of native cover, non-native removal may allow the native community to naturally regenerate following the removal of the physical disturbance. Although adding natives to the landscape is sometimes necessary for restoration, non-native removal alone can be less labor-intensive and more cost-effective [15–18]. Land managers frequently spend a large part of their budget on non-native removal [8], yet the underlying mechanisms and site conditions that allow for the success of this approach are not quantified.

In disturbed areas, non-native annual plants can germinate quickly and competitively exclude native plants from fundamental resources, such as light, nutrients, space, and water [12,19–23]. Ultimately, insufficient access to resources due to competition can reduce native cover, growth, and species richness [19,20,23,24]. In Southern California, non-native annuals are expected to compete more with native shrubs during the seedling establishment phase when they are accessing resources from the same top few centimeters of soil. Non-native annuals leave behind a dense layer of litter and thatch that can reduce the germination of native plants [25–27]. Once their deep roots are established, native shrubs are better able to compete with non-native annuals [7]. Therefore, the removal of non-native species followed by natural regeneration of the native shrub community should reverse the competitive effects of non-native annuals. In fact, several studies have shown that non-native removal successfully increased native cover, density, growth, and species richness, and improved access to light and soil nutrients [28,29]. Removing one non-native can lead to an increase in another non-native, pointing to the importance of evaluating site conditions best suited to non-native removal without active restoration [30,31].

Due to differences in environmental conditions, such as moisture availability and temperature, abundance of non-native species and restoration success may vary between regions [32,33]. Sites located further from the coast experience warmer summer temperatures and greater evapotranspirative stress, which impacts plant species richness and composition [32]. Higher resource environments tend to be more heavily invaded, as most non-natives are fast-growing and able to quickly take advantage of available resources [34–36]. However, non-native species also successfully invade low-resource environments, possibly due to greater efficiency of resource use [37,38]. The native herbaceous annuals (forbs) that comprise the understory of a healthy coastal sage scrub community germinate more in wet years, so regional differences may also influence native species richness by altering the density and diversity of native annuals [39]. This may impact non-native density as well, as diverse native communities are typically more resistant to invasion [40].

Theoretical work suggests that thresholds of resilience may facilitate recovery at sites with high native cover, while heavily degraded areas may not recover [11,41,42]. The existing vegetation cover predicts the species composition of the seed dispersed and vegetation cover presented in the following year [43,44]. Limited native seed rain and profuse non-native seed production, also known as "seed swamping", hinders native recovery [43,45,46]. Areas with high non-native cover often see a reduction in native cover, with sites containing a level of non-native cover exceeding a threshold experiencing greater difficulty with native recovery [24]. In such heavily degraded areas, removing one non-native species without adding natives will typically lead to the invasion of a different non-native species [17]. Less studied is whether there is a management threshold of native cover required for non-native removal without the addition of native plants to be successful.

Previous work at our study site demonstrated that non-native plant removal resulted in higher native cover in weeded plots [47]. In this study, we examined the possible mechanisms and identified necessary site conditions for this result by investigating the relative role of native shrub and forb seedling recruitment versus growth from mature shrubs in areas with varying initial native cover. Specifically, we asked the following: (1) Does non-native plant removal increase native cover through increased seedling recruitment or by promoting the growth of established native shrubs? We hypothesized that removal of non-natives would promote seedling recruitment of native shrubs and forbs by decreasing competition for water and other resources [7,48]. In contrast, we hypothesized that growth of existing shrubs would not be influenced by non-native removal because the perennial natives access deeper water than that used by non-native annuals [49,50]. (2) Does the success of non-native removal vary depending on site conditions? We hypothesized that non-native removal would vary from the coast to inland sites, along with variation in environmental conditions including temperatures, soil nutrients, and soil water-holding capacity [32]. We also hypothesized that we would identify a threshold of initial native cover at which non-native removal was no longer effective [51]. Ultimately, the results of this study reveal the mechanisms by which non-native removal increases native cover and clarify the ideal site conditions for which this technique is most effective.

## 2. Materials and Methods

We established thirteen restoration sites throughout the Nature Reserve of Orange County, California, with ten established in 2010 and three added in 2014 (Figure 1). All 13 study sites were located within Irvine Ranch Natural Landmarks with 6 sites in the inland region and 7 sites in the coastal portion (Figure 1). Study sites were located within invaded areas of coastal sage scrub plant communities ranging in levels of remaining native cover and shared dominant species typical of this community, such as *Artemisia californica*, *Eriogonum fasciculatum*, and *Salvia mellifera* [52]. Non-native species included *Bromus diandrus*, *B. madritensis*, *Avena barbata*, *A. fatua*, *Erodium cicutarium*, *Brassica nigra*, *Hirschfeldia incana*, and *Centauria melitensis*. These are all commonly found within disturbed coastal sage scrub communities and are known to vary in terms of their environmental impact [53,54]. For example, *Brassica nigra* and *Hirschfeldia incana* are known to release allelopathic chemicals, reducing soil microbe abundance and leaving lasting impacts [55,56]. *Erodium cicutarium* and *Bromus madritensis* are commonly found even in native-dominated California plant communities and only displace native shrubs when combined with disturbances, such as drought or increases in fire frequency [57].

Degradation of this plant community in this region is the result of physical disturbance from decades of overgrazing in combination with drought, urban development, and fire, which facilitated invasion by non-native annual grasses and forbs [4,58]. Restoration sites were located on soils ranging from clay loams to sandy loams (Table A1; [59]). Although there was some variation in soil type, slope, and aspect between sites and region, the variation was within the range expected by this habitat type [60]. The climate is Mediterranean, characterized by mild, wet winters and hot, dry summers. The mean annual precipitation

is 32.3 cm (1987–2018 weather, Tustin Irvine Ranch Station, NCDC #9087, 33°44′ N, 117°47′ W, elevation 36 m) with rain falling predominantly from November to April.

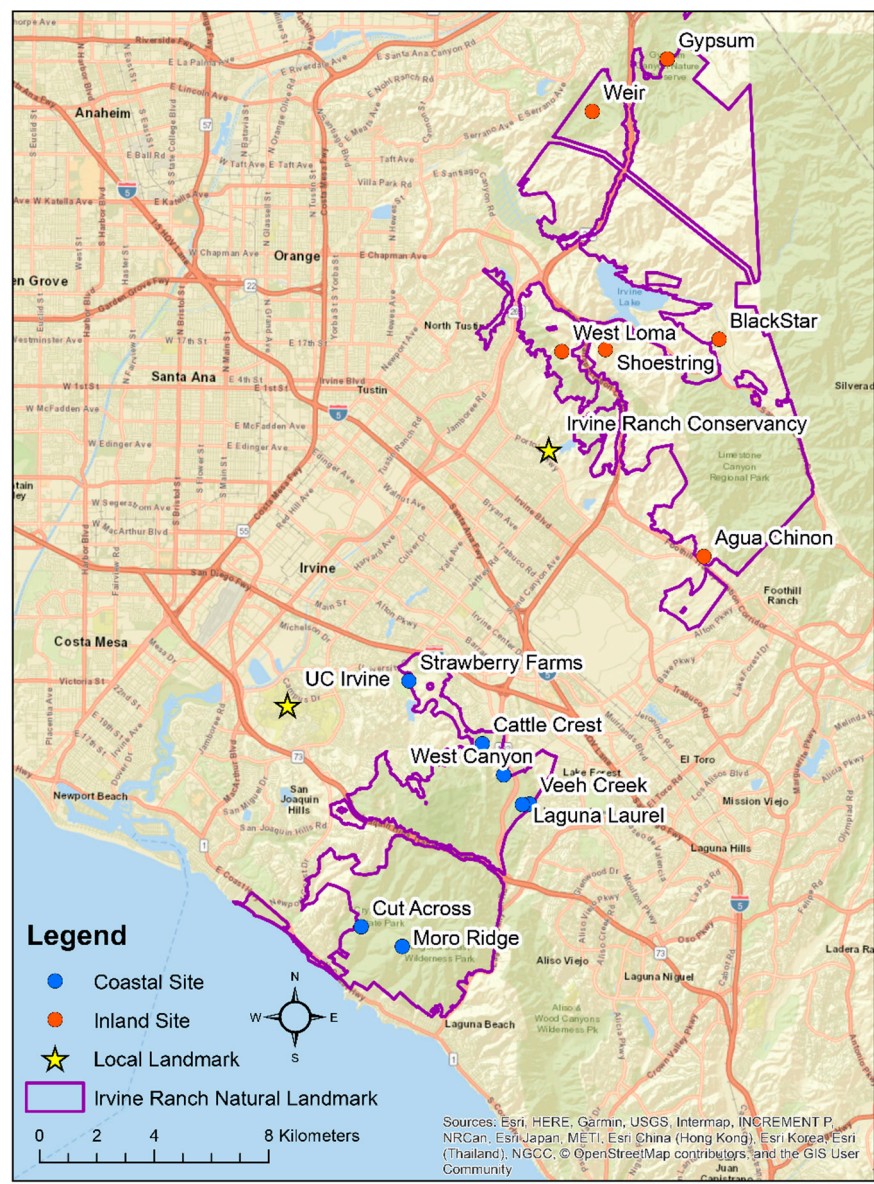

**Figure 1.** Map of study sites. All sites are located within Irvine Ranch Natural Landmarks in Orange County, CA, USA with ten sites (Agua Chinon, Blackstar, Cattle Crest, Laguna Laurel, Shoestring, Strawberry Farms, Weir, West Canyon, West Loma, and Veeh Creek) originally established in 2010 and three sites (Cut Across, Gypsum, and Moro Ridge) added later in 2014.

We selected sites in areas consisting of coastal sage scrub using available digital vegetation maps for the regions [61]. Site selection was finalized after ground-based reconnaissance of candidate locations to ensure they met the criteria for co-dominant species and a range of native shrub cover invaded by non-native annuals (Figure 2). Although native grasses and forbs were also present at the site, the proportion of total vegetation cover that both growth forms occupied were negligible compared to that of native shrubs. Given our interest in identifying thresholds of native cover that optimizes non-native removal, we established four initial native plant cover classes, namely (1) 20–29% low, (2) 30–39% medium-low, (3) 40–49% medium-high, and (4) 50–59% high cover, based on visual estimates. The remaining cover was a mixture of non-native species, bare ground, and negligible amounts of native annuals. Overall plant cover was determined from visual estimates of species-by-species cover taken from the entire plot,

validated with point-intercept estimates subsampled within Griffoul 2017. For each cover class, two 5 m × 5 m plots were established within 10 m of each other. Of these two plots, one plot was randomly assigned to the non-native weed removal treatment while the other became the unmanipulated control. This resulted in a total of 8 experimental plots at each site, with each plot representing a unique native cover class and weed treatment combination. In 2014, we installed a smaller 0.5 m × 0.5 m subplot in a random, open portion of the 5 m × 5 m plot to obtain representative assessments of native seedling recruitment.

(a)  (c)

(b)

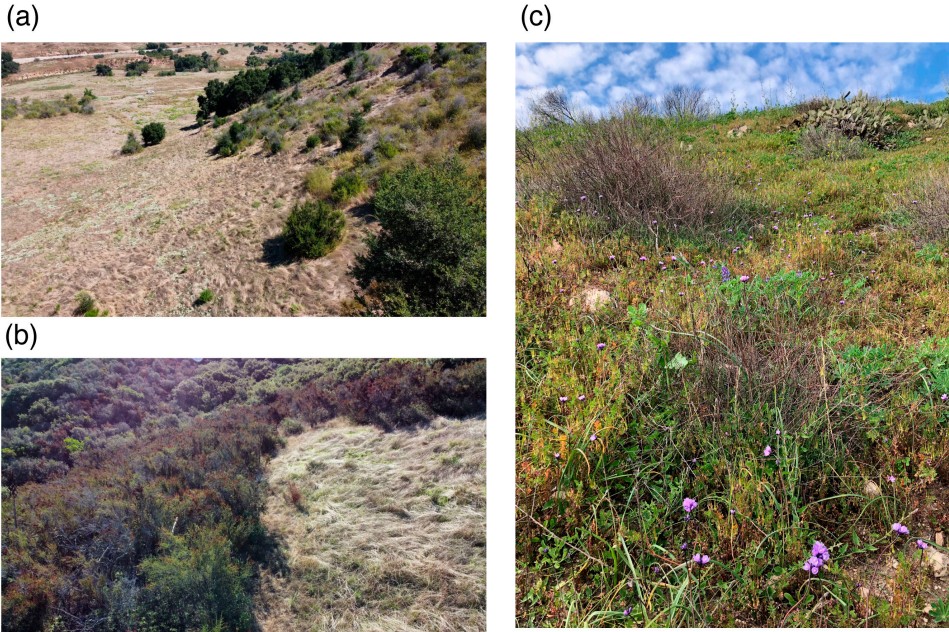

**Figure 2.** Photographs showing (**a**) non-native annual grassland with some coastal sage scrub interspersed, (**b**) coastal sage scrub adjacent to non-native grassland, and (**c**) a close-up of the Shoestring Canyon site, showing native and non-native species interspersed. Photos by Kristin Barbour (**a**,**c**) and Priscilla Ta (**c**).

We started removing non-native species at the initiation of the study in 2010. At the beginning of each growing season, we applied low-dose glyphosate, a broad-spectrum herbicide, at 0.5 qts/ac (1.2 L/ha) for a 0.5% concentration of Roundup PROMAX® using a backpack sprayer to favor native germination by eliminating non-native species that sprouted after the winter rains. The start of each growing season was marked by the first germinating rain events of the year that typically occur around January. In situations where non-natives were too interspersed with native species for us to safely spot-treat with herbicide, we waited until later in the season to remove the non-natives by hand. Every May, we mechanically removed non-native plants by hand or with weed-eaters to further reduce competition and the quantity of non-native seed entering the seed bank. We implemented the weed removal treatment within a 7 × 7 m area (5 × 5 m plot with 1 m extensions). Non-native removal was conducted within each weed-treated plot, specifically targeting areas densely covered with non-natives to mimic actual weeding methods used in local large-scale restoration projects that reduce non-natives without removing every single non-native plant. Litter and thatch were not intentionally removed along with the non-natives, although weeded plots did have a significantly greater cover of bare ground compared to control plots across all years ($F_{1,252}$ = 18.92, *p* = 0.0001).

Every year, from 2014 to 2016, we collected plant density measurements at all thirteen restoration sites to study how non-native removal impacted seedling recruitment throughout the growing season. Near the beginning of each growing season, in January, all native and non-native seedlings were identified and counted within the smaller 0.5 × 0.5 m subplots. We defined seedlings as young plants that have germinated and emerged following the first winter

rain event of the year. In May, we recorded the density of native and non-native individuals still present in our subplot towards the end of the growing season.

To evaluate how non-native plant removal affected established shrub growth, we measured and calculated the volume of all native shrubs taller than 1 m within each experimental plot in fall 2012, towards the beginning of the recent severe drought in California, and again in Spring 2014, the peak year of the drought. During fall 2012, we only collected data at the ten restoration sites that were established in 2010 and which had already received two full years of weed removal. We did not sample the three restoration sites (Cut Across, Moro Ridge, and Gypsum) that were added in 2014. During spring 2014, we collected shrub data at all thirteen restoration sites. Size measurements, taken at the base of the shrub, included height (H), length at the widest point (L), and width at a 90º angle from the widest point (W). Shrub volume, assumed to be cylindrical, was calculated using the formula $V_{shrub} = \pi \times ((W + L)/4)^2 \times H$. The average shrub volume, total shrub volume, and number of shrubs were calculated for each plot and year of measurement. In 2014, many dead shrubs were observed in our plots, so the number of dead shrubs was also recorded.

To evaluate the effect of non-native plant removal on native and non-native plant density and species richness over the course of three years (2014–2016) for winter and spring, we performed repeated measures mixed model ANOVAs followed by Tukey–Kramer post hoc tests using the PROC MIXED SAS 9.4 software [62]. Plot was included in the model as the repeated factor (measured in multiple years) while treatment (control or weeded), initial native cover (low, medium-low, medium-high, or high), and region (coastal or inland) were fixed factors. Site (the thirteen study locations, an experimental blocking unit containing all treatment combinations) was included as a random factor in our analyses. Plant density data were log-transformed ($\ln(x + 1)$) while species richness data were square-root-transformed ($\sqrt{x + 1}$) to satisfy ANOVA's requirement for normality. We performed these analyses on the total density of native, native shrub, native forb, and non-native seedlings, and on native and non-native species richness.

To study the effect of non-native removal on already-established native shrubs, we again used repeated measures mixed model ANOVAs to compare differences in the average volume per plot, total volume per plot, and the number of living native shrubs at the ten restoration sites originally established in 2010. We used the same model described above (treatment, initial native cover, and region as fixed factors, site as a random factor, and plot as the repeated factor) to analyze shrub size and number. We excluded the three sites (Cut Across, Moro Ridge, and Gypsum) that were established in 2014 because shrub size data was only collected once in 2014. Average shrub volume and number were both ln-transformed to fit a normal distribution. We used PROC GLIMMIX to conduct a logistic regression with site as a random variable and treatment, region, and cover as fixed factors, to determine whether the proportion of dead native shrubs observed in 2014 was related to treatment, initial native cover, or region.

## 3. Results

Non-native removal had mixed effects on native coastal sage scrub seedling recruitment. Although non-native removal did not increase the density of native seedlings that germinated in the winter, native density was significantly higher in weeded plots in the spring (Tables 1 and A2, Figure 3a). When evaluating the composition of the native seedlings based on functional group, we found that shrub seedlings were significantly more abundant in weeded compared to control plots in winter and spring, while herbaceous forb species were not influenced by non-native removal in either season (Tables 1 and A2, Figure 3b,c). Non-native removal significantly reduced non-native seedling recruitment, with the density of non-natives germinating in the winter and present in the spring consistently lower in weeded plots (Tables 1 and A2, Figure 3d).

**Table 1.** Results from repeated measures mixed model ANOVA for native, native shrub, native forb, and non-native density. Treatment refers to whether plots experienced non-native plant removal (weeded) or not (control). Native cover class refers to the initial range of native cover for the plot (20–29% low, 30–39% medium-low, 40–49% medium-high, 50–59% high). Region refers to whether the restoration site was in the coastal or inland portion of Irvine Ranch Natural Landmarks. Shaded rows highlight effects that were significant ($p < 0.05$). Data were ln(x + 1)-transformed.

| Response Variable | Season | Effect | Num DF | Den DF | F Value | *p* Value |
|---|---|---|---|---|---|---|
| Native density | Winter | Treatment | 1 | 285 | 0.25 | 0.6198 |
| | | Native cover class | 3 | 285 | 0.19 | 0.9014 |
| | | Native cover class × treatment | 3 | 285 | 1.13 | 0.3374 |
| | | Region | 1 | 285 | 3.78 | 0.0527 |
| | | Region × treatment | 1 | 285 | 1.96 | 0.1623 |
| | | Region × native cover class | 3 | 285 | 0.84 | 0.4713 |
| | | Region × native cover class × treatment | 3 | 285 | 0.82 | 0.4856 |
| | Spring | Treatment | 1 | 283 | 7.57 | 0.0063 |
| | | Native cover class | 3 | 283 | 0.39 | 0.7583 |
| | | Native cover class × treatment | 3 | 283 | 4.65 | 0.0035 |
| | | Region | 1 | 283 | 14.97 | 0.0001 |
| | | Region × treatment | 1 | 283 | 1.59 | 0.2081 |
| | | Region × native cover class | 3 | 283 | 1.47 | 0.224 |
| | | Region × native cover class × treatment | 3 | 283 | 0.41 | 0.7439 |
| Native shrub density | Winter | Treatment | 1 | 285 | 6.73 | 0.01 |
| | | Native cover class | 3 | 285 | 2.12 | 0.098 |
| | | Native cover class × treatment | 3 | 285 | 1.32 | 0.2669 |
| | | Region | 1 | 285 | 1.43 | 0.233 |
| | | Region × treatment | 1 | 285 | 0.79 | 0.3743 |
| | | Region × native cover class | 3 | 285 | 0.33 | 0.8033 |
| | | Region × native cover class × treatment | 3 | 285 | 2.26 | 0.0818 |
| | Spring | Treatment | 1 | 283 | 10.86 | 0.0011 |
| | | Native cover class | 3 | 283 | 0.36 | 0.7832 |
| | | Native cover class × treatment | 3 | 283 | 1.14 | 0.332 |
| | | Region | 1 | 283 | 4.5 | 0.0347 |
| | | Region × treatment | 1 | 283 | 0.12 | 0.7328 |
| | | Region × native cover class | 3 | 283 | 0.12 | 0.9472 |
| | | Region × native cover class × treatment | 3 | 283 | 0.77 | 0.5101 |
| Native forb density | Winter | Treatment | 1 | 285 | 0.47 | 0.493 |
| | | Native cover class | 3 | 285 | 0.44 | 0.727 |
| | | Native cover class × treatment | 3 | 285 | 1.09 | 0.3529 |
| | | Region | 1 | 285 | 4.69 | 0.0311 |
| | | Region × treatment | 1 | 285 | 0.14 | 0.7105 |
| | | Region × native cover class | 3 | 285 | 1.4 | 0.1583 |
| | | Region × native cover class × treatment | 3 | 285 | 0.6 | 0.6148 |
| | Spring | Treatment | 1 | 283 | 0.01 | 0.9049 |
| | | Native cover class | 3 | 283 | 1.12 | 0.3428 |
| | | Native cover class × treatment | 3 | 283 | 1.6 | 0.1889 |
| | | Region | 1 | 283 | 12.8 | 0.0004 |
| | | Region × treatment | 1 | 283 | 1.88 | 0.1718 |
| | | Region × native cover class | 3 | 283 | 2.91 | 0.0351 |
| | | Region × native cover class × treatment | 3 | 283 | 1.53 | 0.2074 |

**Table 1.** *Cont.*

| Response Variable | Season | Effect | Num DF | Den DF | F Value | *p* Value |
|---|---|---|---|---|---|---|
| Non-native density | Winter | Treatment | 1 | 285 | 19.21 | <0.0001 |
| | | Native cover class | 3 | 285 | 2.45 | 0.0638 |
| | | Native cover class × treatment | 3 | 285 | 0.78 | 0.5057 |
| | | Region | 1 | 285 | 0.03 | 0.872 |
| | | Region × treatment | 1 | 285 | 1.09 | 0.2967 |
| | | Region × native cover class | 3 | 285 | 0.31 | 0.8198 |
| | | Region × native cover class × treatment | 3 | 285 | 0.19 | 0.903 |
| | Spring | Treatment | 1 | 283 | 22.66 | <0.0001 |
| | | Native cover class | 3 | 283 | 3.73 | 0.0118 |
| | | Native cover class × treatment | 3 | 283 | 1.64 | 0.1804 |
| | | Region | 1 | 283 | 0.05 | 0.816 |
| | | Region × treatment | 1 | 283 | 2.72 | 0.1002 |
| | | Region × native cover class | 3 | 283 | 2.32 | 0.0754 |
| | | Region × native cover class × treatment | 3 | 283 | 0.49 | 0.6888 |

Weeded plots with the two lowest initial cover classes (20–29% low and 30–39% medium-low) contained significantly higher native seedling density compared to control plots with the same level of initial cover, suggesting that non-native removal is most effective in areas with an initial native cover of 20–39% (Tables 1 and A2). The initial native plant cover did not significantly impact native shrub and forb density (Table 1) but did influence non-native seedling density in the spring (Table 1). Non-native density was greatest in plots with 20–29% low and 30–39% medium-low native cover (Table A2).

Native seedling density was higher at inland restoration sites compared to coastal sites throughout the growing season in winter and spring (Tables 1 and A2). Inland sites contained more native shrub and forb seedlings in both seasons (Tables 1 and A2). However, there was no significant difference in non-native density between regions (Table 1).

Native species richness was highest in the spring for weeded plots with 20–29% low and 30–39% medium-low initial native cover, suggesting once again the effectiveness of non-native removal for areas with an initial cover of 20–39% natives (Tables 2 and A3). When evaluating the effect of non-native removal on non-native species richness, we found that non-native species richness was significantly reduced in weeded plots in the winter (Tables 2 and A3). While non-native species richness remained lower in weeded plots in the spring, the difference was insignificant (Table 2).

Native species richness was significantly greater at inland restoration sites, especially in the spring (Table 2). Interestingly, native species richness was greatest in the inland control plots followed by inland weeded, coastal weeded, and coastal control plots (Tables 2 and A3). Non-native species richness was also significantly higher at inland restoration sites in the winter (Table 2). Non-native richness remained significantly higher in the spring for inland plots with an initial native cover ranging from 20–49% (Table 2, Table A3).

Non-native plant removal did not significantly impact established shrub volume but did result in significantly more established shrubs (Tables A4 and A5c). The initial cover of existing native plants did not significantly impact the average volume or number of established shrubs, but the total shrub volume in each plot did vary significantly depending on initial native cover (Table A4). Total shrub volume followed the same pattern of the four initial cover classes, with plots in the highest cover class (50–59% high) having the greatest total volume, while plots in the two lowest cover classes (20–29% low and 30–39% medium-low) had the least (Table A5b). When evaluating the influence of region on established shrub volume and number, we found that inland restoration sites had, on average, significantly smaller shrubs, but significantly more established native shrubs compared to the coastal sites (Figure 4, Tables A4 and A5a–c). Since we observed several dead established shrubs in our plots in 2014, we decided to assess how shrub mortality was affected by non-native removal, initial native cover, and region. We found that shrub

mortality was lower in the weeded plots than in the control plots and lower at coastal sites compared to inland sites (Figure 5, Table A6).

**(a) Plant Density Response to Non-Native Removal - Native Seedlings**

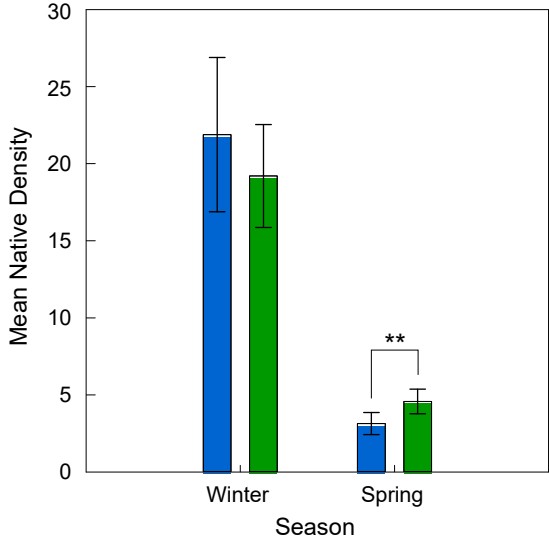

**(b) Plant Density Response to Non-Native Removal - Native Shrub Seedlings**

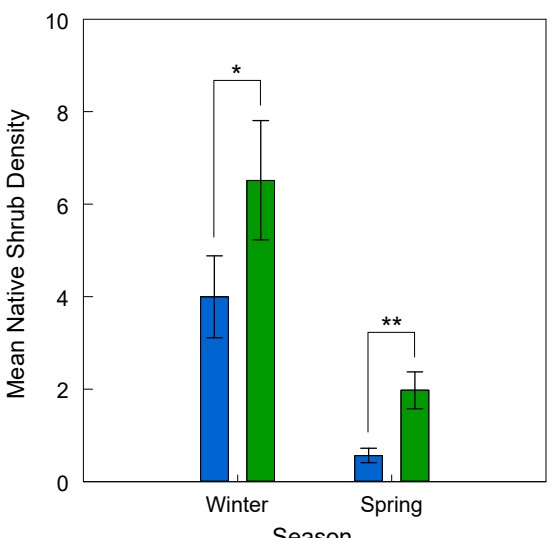

**(c) Plant Density Response to Non-Native Removal - Native Forb Seedlings**

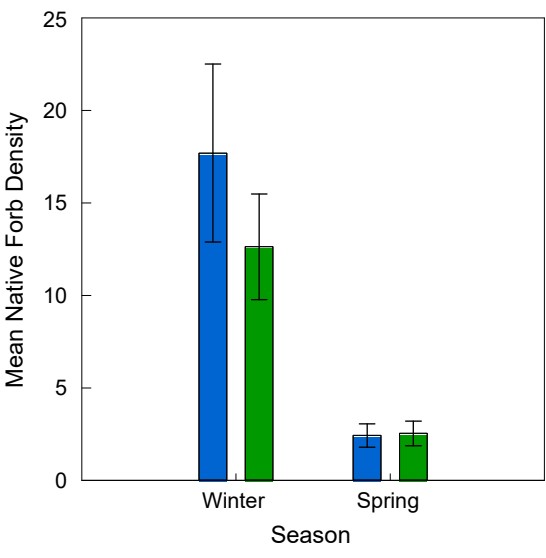

**(d) Plant Density Response to Non-Native Removal - Non-Native Seedlings**

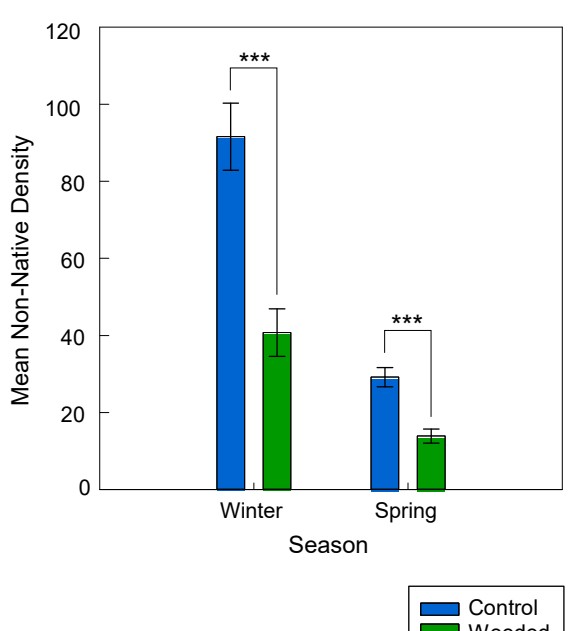

**Figure 3.** ANOVA results on the effects of non-native plant removal on seedling recruitment of (**a**) all natives, (**b**) native shrubs, (**c**) native forbs, and (**d**) all non-natives. Plant density data were collected over the course of a growing season for three years (2014–2016). Results are reported as mean ± SE. Significant factors are included in the graphs with * signifying $p < 0.05$, ** signifying $p < 0.01$, and *** signifying $p < 0.0001$.

**Table 2.** Results from repeated measures mixed model ANOVA for native and non-native species richness. Treatment refers to whether plots experienced non-native plant removal (weeded) or not (control). Native cover class refers to the initial range of native cover for the plot (20–29% low, 30–39% medium-low, 40–49% medium-high, 50–59% high). Region refers to whether the restoration site was in the coastal or inland portion of Irvine Ranch Natural Landmarks. Shaded rows highlight effects that were significant ($p < 0.05$). Data were sqrt (x + 1)-transformed.

| Response Variable | Season | Effect | Num DF | Den DF | F Value | *p* Value |
|---|---|---|---|---|---|---|
| Native species richness | Winter | Treatment | 1 | 285 | 0.01 | 0.9366 |
| | | Native cover class | 3 | 285 | 0.11 | 0.9551 |
| | | Native cover class × treatment | 3 | 285 | 2.21 | 0.0873 |
| | | Region | 1 | 285 | 3.18 | 0.0756 |
| | | Region × treatment | 1 | 285 | 1.48 | 0.2249 |
| | | Region × native cover class | 3 | 285 | 0.38 | 0.7703 |
| | | Region × native cover class × treatment | 3 | 285 | 0.49 | 0.6927 |
| | Spring | Treatment | 1 | 283 | 0.38 | 0.54 |
| | | Native cover class | 3 | 283 | 0.33 | 0.802 |
| | | Native cover class × treatment | 3 | 283 | 8.1 | <0.0001 |
| | | Region | 1 | 283 | 12.38 | 0.0005 |
| | | Region × treatment | 1 | 283 | 8.02 | 0.005 |
| | | Region × native cover class | 3 | 283 | 1.36 | 0.2564 |
| | | Region × native cover class × treatment | 3 | 283 | 0.18 | 0.9104 |
| Non-native species richness | Winter | Treatment | 1 | 285 | 4.85 | 0.0284 |
| | | Native cover class | 3 | 285 | 1.5 | 0.2148 |
| | | Native cover class × treatment | 3 | 285 | 0.51 | 0.6756 |
| | | Region | 1 | 285 | 5.2 | 0.0233 |
| | | Region × treatment | 1 | 285 | 1.17 | 0.2795 |
| | | Region × native cover class | 3 | 285 | 0.83 | 0.4757 |
| | | Region × native cover class × treatment | 3 | 285 | 0.55 | 0.6478 |
| | Spring | Treatment | 1 | 283 | 2.91 | 0.0893 |
| | | Native cover class | 3 | 283 | 2.21 | 0.0874 |
| | | Native cover class × treatment | 3 | 283 | 0.09 | 0.9638 |
| | | Region | 1 | 283 | 1.88 | 0.172 |
| | | Region × treatment | 1 | 283 | 0.37 | 0.5426 |
| | | Region × native cover class | 3 | 283 | 2.77 | 0.042 |
| | | Region × native cover class × treatment | 3 | 283 | 1.18 | 0.3186 |

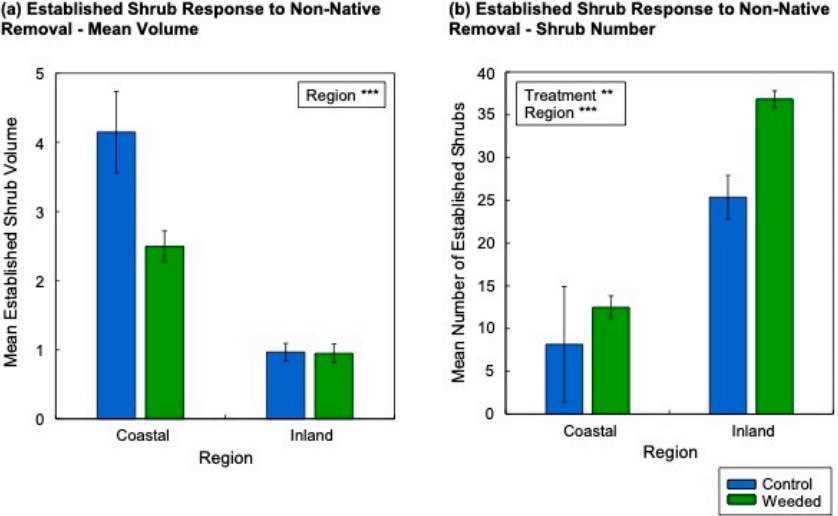

**Figure 4.** ANOVA results on the effects of non-native plant removal across region on (**a**) mean volume and (**b**) number of established native shrubs. Shrub data were collected in 2012 and 2014 at the ten original restoration sites. Results are reported as mean ± SE. Significant main effects are reported in the text box with ** indicating $p < 0.01$, and *** indicating $p < 0.0001$. The treatment-by-region interaction was not significant for either analysis and so was not listed.

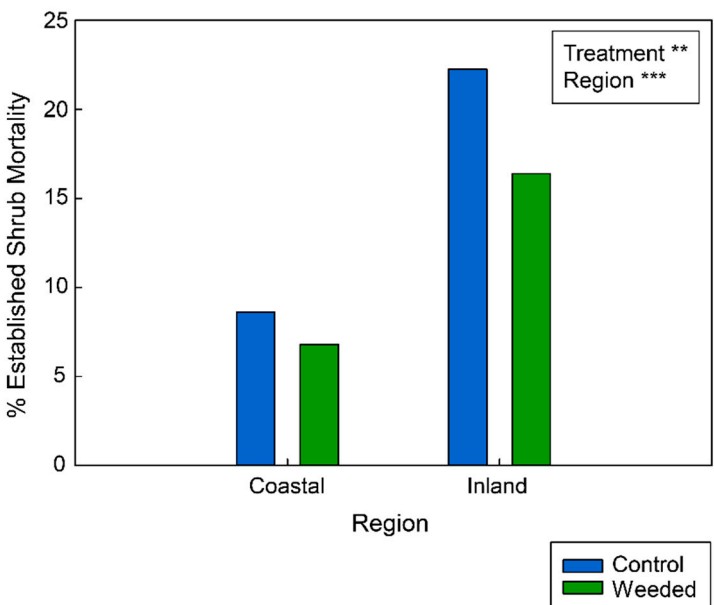

**Figure 5.** The effect of non-native plant removal across region on established native shrub mortality. Shrub mortality was assessed once in 2014 for all thirteen restoration sites. Results are expressed as a percentage of dead shrubs. Significance levels are marked by asterisks, with ** representing $p < 0.01$, and *** representing $p < 0.0001$. The treatment-by-region interaction was included in the model but was not significant ($p > 0.05$).

## 4. Discussion

Non-native removal in areas with existing native cover was successful, but our results revealed some surprising mechanisms behind its success. First, our hypothesis that non-native control would increase native density and species richness was not supported in the early growing season, but higher native density and richness was observed at the end of the growing season, indicating greater native survival in weeded plots. Secondly, non-native removal did not increase established native shrub volume as expected but did increase the number of established shrubs present and reduced mortality, something that was also found in a post-fire non-native removal study [63]. Thirdly, our hypothesis that the effects of non-native removal would vary with site conditions was supported, as removal was more effective inland than at coastal sites. Finally, we identified a threshold of initial native cover (20–39%) for which non-native removal was most effective. The removal of non-natives in areas with higher initial native cover was less effective at increasing the number and diversity of natives, indicating that land stewards could focus removal efforts on the areas with 20–39% native cover to achieve the greatest impact.

Non-native plant removal has been shown to increase native cover and species richness, but only in areas with existing natives that can naturally recolonize weeded areas through native seed dispersal and germination or clonal recruitment [64–67]. With the removal of non-native species, competition for resources, such as water, space, and light, that are crucial during the early stages of native plant establishment, are reduced [12,20,21,68,69]. Indeed, we found native seedling density and species richness to be higher in weeded plots—something supported by other studies [70,71]. This effect of non-native removal on plant density was most pronounced for native shrub seedlings, which were denser in weeded areas, as opposed to forbs which remained the same. Non-native plants are not always harmful to native plants and may even provide ecosystem services without outcompeting natives [72–74]. Low numbers of native forb seedlings might indicate an inadequate seedbank and need to add seeds [7,75,76]. Our method of non-native removal, which included chemical and mechanical control measures, may have had an unintended impact on native forb seedlings. Herbicide residuals or spillovers can reduce recruitment of non-target species, such as native forbs [77]. Mechanical removal often disturbs the soil

which may trigger further non-native establishment and negatively impact root systems and the mycorrhizal fungi of the remaining native plants, subsequently affecting plant growth [78–80]. The prolonged drought occurring during our study period likely impacted native forb recruitment in both control and weeded plots since forbs are typically more abundant in wet years [39,52].

Contrary to prior studies showing a positive impact of non-native removal on native shrub growth [23,29], non-native removal did not increase established shrub volume in our study. Weeding may not have had an effect because established shrubs can access nutrients and water from deeper depths, ultimately making them better competitors against non-native annuals [81]. Therefore, it is possible that other factors, such as the recent severe drought, may have inhibited shrub growth. Although coastal sage scrub shrub species are adapted to drought, prolonged drought can negatively impact native biomass [82,83]. For example, Llorens et al. (2004) found that stem elongation and shoot growth of mature shrubs was strongly reduced under drought conditions [84]. Native shrubs were larger, on average, in the coastal regions of our study, possibly due to greater moisture near the coast or lower fire frequency in coastal areas [52].

Despite having no effect on established shrub volume, non-native removal significantly increased the number of established shrubs and reduced shrub mortality. The positive impact of non-native removal on established shrub number suggests its importance for reducing competition for resources that would allow germinated shrub seedlings to reach maturity. Non-native plants often grow faster and are better at capitalizing on available resources, ultimately outcompeting native plants and limiting native seedling survival and growth [18,85]. Increased resource availability to native plants following non-native removal may have also contributed to the reduction in shrub mortality in weeded plots. Competition with non-native plants can increase vulnerability of established shrubs to die-off, especially in conditions of high resource stress, such as drought [86]. The drier conditions of inland sites likely exacerbated the influence of drought, possibly contributing to the greater shrub die-off observed inland [87].

We identified 40% as a threshold of native cover above which non-native removal may be less effective. Native density and species richness were highest in weeded plots with lower initial native cover, demonstrating the effectiveness of non-native removal for areas with 20–39% existing native cover. For plots with an initial native cover beyond 39%, there was no significant positive effect of non-native removal on native seedling density and richness. Theoretical work suggests there may be thresholds of native cover beyond which communities may be stuck in a degraded state and not recover [42]. In such degraded sites, removing non-native species without adding natives will simply allow one non-native to replace another [45,88,89]. We did not test non-native removal in areas with less than 20% cover because existing studies indicate these areas would be unlikely to recover. However, our results suggest that a management threshold exists where non-native removal is not as effective for areas with greater than 39% initial native cover. One possible explanation is that areas with higher initial native cover resulted in less available resources for new natives to become established, especially during the drought. Our result indicates the importance of considering site conditions prior to weed removal efforts. Other studies have documented the possibility of harming native communities through weed removal [14,90,91]. Depending on site conditions, limited management resources may be better directed towards other activities [8].

Our study revealed that non-native removal in heavily degraded coastal sage scrub communities increased native cover through increased native seedling recruitment rather than through established shrub growth. Non-native plant removal in degraded areas also helped to support native shrub density, species richness, established shrub number, and to reduce shrub mortality. Climate models predict that our region will experience increasingly severe and common droughts along with higher temperatures [92]. Reducing competition with non-natives may be critical to the survival of a diverse native coastal sage scrub community, and we recommend this approach for transitional areas with 20–39%

of existing native cover. Indeed, our finding that non-native removal was not effective in areas with more than 40% shrub cover indicates that these areas would be best left undisturbed by removal efforts. In conclusion, our results indicate the effectiveness of non-native removal for increasing native cover and diversity in areas with a low to medium cover of native species.

**Author Contributions:** J.C.B., M.L., Q.S., I.O., S.K. and T.E.H. designed and implemented the experiment; E.G., K.R.B., K.T.S., P.M.T., Q.S. and I.O. collected the data; P.M.T. and S.K. analyzed the data; P.M.T., S.K., K.T.S. and J.J.L. wrote the manuscript; all authors edited the manuscript. All authors have read and agreed to the published version of the manuscript.

**Funding:** This research was funded by the Center for Environmental Biology, which receives support from the Natural Communities Coalition and The Nature Conservancy.

**Institutional Review Board Statement:** Not applicable.

**Data Availability Statement:** The data presented in this study are available in Dryad at https://doi.org/10.7280/D1168Q.

**Acknowledgments:** Thanks to staff at the Irvine Ranch Conservancy for helping establish and maintain the experimental plots. Additional thanks to all staff and undergraduate interns from the University of California, Irvine's Center for Environmental Biology, for their support and assistance with data collection. Finally, thanks to the Nature Conservancy and the Natural Communities Coalition.

**Conflicts of Interest:** The authors declare no conflict of interest.

## Appendix A  Soil Composition of Study Sites

**Table A1.** Soil composition of assisted passive restoration sites. This table presents the soil composition of each restoration site.

| Region | Restoration Site | Soil Composition |
|---|---|---|
| Coastal | Cattle Crest | 71.3% Anaheim loam, 28.7% Cieneba-Rock outcrop complex |
| Coastal | Cut Across | 92.4% Bosanko clay, 7.6 Cieneba sandy loam |
| Coastal | Laguna Laurel | 100% Yorba cobbly sandy loam |
| Coastal | Moro Ridge | 100% Myford sandy loam |
| Coastal | Strawberry Farms | 100% Alo clay |
| Coastal | Veeh Creek | 87.9% Alo variant clay, 6.4% Capistrano sandy loam, 5.8% Yorba cobbly sandy loam |
| Coastal | West Canyon | 97.6% Cieneba sandy loam, 2.4% Myford sandy loam |
| Inland | Agua Chinon | 100% Balcom clay loam |
| Inland | Blackstar | 100% Soper gravelly loam |
| Inland | Gypsum | 100% Cieneba-Rock outcrop complex |
| Inland | Shoestring | 100% Calleguas clay loam |
| Inland | Upper Weir | 100% Alo clay |
| Inland | West Loma | 100% Anaheim loam |

## Appendix B Tukey Post Hoc Results for Native and Non-Native Seedling Density

**Table A2.** Tukey post hoc results for native and non-native seedling density. Tukey post hoc results for native density, native shrub density, native forb density, and non-native density on the significant effects of non-native plant removal treatment, native cover class, region, and interactions in the winter and spring. Treatment refers to whether plots experienced non-native plant removal (weeded) or not (control). Native cover class refers to the initial range of native cover for the plot (20–29% low, 30–39% medium-low, 40–49% medium-high, 50–59% high). Region refers to whether the restoration site was in the coastal or inland portion of Irvine Ranch Natural Landmarks. Data were ln(x + 1)-transformed. Standard error (SE) and letters from the Tukey post hoc tests are included in results tables, with different letters indicating significant differences and shared letters indicating no significant difference amongst groups.

| Effect | Dependent Variable | Season | Treatment | Estimate | SE | Letter |
|---|---|---|---|---|---|---|
| Treatment | ln(x + 1) native density | Winter | Weeded | 1.6622 | 0.2446 | A |
| | | Winter | Control | 1.5918 | 0.2446 | A |
| | | Spring | Weeded | 0.9912 | 0.133 | A |
| | | Spring | Control | 0.7566 | 0.133 | B |

| Effect | Dependent Variable | Season | Native Cover Class | Treatment | Estimate | SE | Letter |
|---|---|---|---|---|---|---|---|
| Treatment x native cover class | ln(x + 1) native density | Winter | Medium-low | Weeded | 1.8268 | 0.3 | A |
| | | Winter | High | Control | 1.7696 | 0.3 | A |
| | | Winter | Low | Weeded | 1.7183 | 0.3 | A |
| | | Winter | Medium-high | Control | 1.6833 | 0.3 | A |
| | | Winter | High | Weeded | 1.6571 | 0.3 | A |
| | | Winter | Low | Control | 1.5132 | 0.3 | A |
| | | Winter | Medium-high | Weeded | 1.4464 | 0.3 | A |
| | | Winter | Medium-low | Control | 1.401 | 0.3 | A |
| | | Spring | Medium-low | Weeded | 1.2168 | 0.1687 | A |
| | | Spring | Low | Weeded | 1.144 | 0.1687 | AB |
| | | Spring | High | Control | 0.9414 | 0.1687 | ABC |
| | | Spring | Medium-high | Weeded | 0.8223 | 0.1703 | ABC |
| | | Spring | Medium-high | Control | 0.8185 | 0.1687 | ABC |
| | | Spring | High | Weeded | 0.7817 | 0.1687 | ABC |
| | | Spring | Medium-low | Control | 0.68 | 0.1703 | BC |
| | | Spring | Low | Control | 0.5865 | 0.1687 | C |

| Effect | Dependent Variable | Season | Region | Estimate | SE | Letter |
|---|---|---|---|---|---|---|
| Region | ln(x + 1) native density | Winter | Inland | 2.0823 | 0.3436 | A |
| | | Winter | Coastal | 1.1716 | 0.3181 | A |
| | | Spring | Inland | 1.3612 | 0.1849 | A |
| | | Spring | Coastal | 0.3866 | 0.171 | B |

| Effect | Dependent Variable | Season | Treatment | Estimate | SE | Letter |
|---|---|---|---|---|---|---|
| Treatment | ln(x + 1) native shrub density | Winter | Weeded | 1.0694 | 0.1523 | A |
| | | Winter | Control | 0.8016 | 0.1523 | B |
| | | Spring | Weeded | 0.5142 | 0.08285 | A |
| | | Spring | Control | 0.219 | 0.08285 | B |

| Effect | Dependent Variable | Season | Region | Estimate | SE | Letter |
|---|---|---|---|---|---|---|
| Region | ln(x + 1) native shrub density | Winter | Inland | 1.1067 | 0.2102 | A |
| | | Winter | Coastal | 0.7643 | 0.1946 | A |
| | | Spring | Inland | 0.5145 | 0.1024 | A |
| | | Spring | Coastal | 0.2187 | 0.09457 | B |

| Effect | Dependent Variable | Season | Region | Estimate | SE | Letter |
|---|---|---|---|---|---|---|
| Region | ln(x + 1) native forb density | Winter | Inland | 1.6502 | 0.3158 | A |
| | | Winter | Coastal | 0.7178 | 0.2924 | B |
| | | Spring | Inland | 0.9825 | 0.169 | A |
| | | Spring | Coastal | 0.1589 | 0.1563 | B |

| Effect | Dependent Variable | Season | Region | Native Cover Class | Estimate | SE | Letter |
|---|---|---|---|---|---|---|---|
| Interaction between region and initial native cover class | ln(x + 1) native forb density | Winter | Inland | High | 1.8096 | 0.3654 | A |
| | | Winter | Inland | Medium-low | 1.7847 | 0.3654 | A |
| | | Winter | Inland | Low | 1.5315 | 0.3654 | A |
| | | Winter | Inland | Medium-high | 1.475 | 0.3654 | A |
| | | Winter | Coastal | Low | 0.9536 | 0.3383 | A |
| | | Winter | Coastal | Medium-high | 0.7928 | 0.3383 | A |
| | | Winter | Coastal | High | 0.7569 | 0.3383 | A |

**Table A2.** *Cont.*

| Effect | Dependent Variable | Season | Region | Native Cover Class | Estimate | SE | Letter |
|---|---|---|---|---|---|---|---|
| | | Winter | Coastal | Medium-low | 0.3681 | 0.3383 | A |
| | | Spring | Inland | Medium-low | 1.2256 | 0.1894 | A |
| | | Spring | Inland | High | 0.9492 | 0.1884 | AB |
| | | Spring | Inland | Medium-high | 0.9221 | 0.1894 | AB |
| | | Spring | Inland | Low | 0.8332 | 0.1884 | ABC |
| | | Spring | Coastal | Medium-high | 0.2384 | 0.1744 | BC |
| | | Spring | Coastal | High | 0.2137 | 0.1744 | BC |
| | | Spring | Coastal | Low | 0.1195 | 0.1744 | C |
| | | Spring | Coastal | Medium-low | 0.06407 | 0.1744 | C |

| Effect | Dependent Variable | Season | Treatment | Estimate | SE | Letter |
|---|---|---|---|---|---|---|
| Treatment | ln(x + 1) non-native density | Winter | Control | 3.6099 | 0.2253 | A |
| | | Winter | Weeded | 2.6637 | 0.2253 | B |
| | | Spring | Control | 2.6157 | 0.207 | A |
| | | Spring | Weeded | 1.8246 | 0.207 | B |

| Effect | Dependent Variable | Season | Native Cover Class | Estimate | SE | Letter |
|---|---|---|---|---|---|---|
| Initial native cover class | ln(x + 1) non-native density | Winter | Medium-low | 3.5293 | 0.2721 | A |
| | | Winter | Low | 3.3051 | 0.2721 | A |
| | | Winter | High | 2.8946 | 0.2721 | A |
| | | Winter | Medium-high | 2.8182 | 0.2721 | A |
| | | Spring | Medium-low | 2.5256 | 0.2384 | A |
| | | Spring | Low | 2.4608 | 0.2376 | AB |
| | | Spring | High | 2.015 | 0.2376 | AB |
| | | Spring | Medium-high | 1.8791 | 0.2384 | B |

## Appendix C  Tukey Post Hoc Results for Native and Non-Native Species Richness

**Table A3.** Tukey post hoc results for native and non-native species richness. Results for native and non-native species richness on the significant effects of non-native plant removal treatment, native cover class, region, and interactions in the winter and spring. Treatment refers to whether plots experienced non-native plant removal (weeded) or not (control). Native cover class refers to the initial range of native cover for the plot (20–29% low, 30–39% medium-low, 40–49% medium-high, 50–59% high). Region refers to whether the restoration site was in the coastal or inland portion of Irvine Ranch Natural Landmarks. Data were sqrt (x + 1)-transformed. Standard error (SE) and letters from the Tukey post hoc tests are included in results tables, with different letters indicating significant differences and shared letters indicating no significant difference amongst groups.

| Effect | Dependent Variable | Season | Native Cover Class | Treatment | Estimate | SE | Letter |
|---|---|---|---|---|---|---|---|
| Treatment x initial native cover class | sqrt (x + 1) native species richness | Winter | High | Control | 1.7302 | 0.1206 | A |
| | | Winter | Medium-low | Weeded | 1.6614 | 0.1206 | A |
| | | Winter | Low | Weeded | 1.645 | 0.1206 | A |
| | | Winter | Medium-high | Control | 1.6201 | 0.1206 | A |
| | | Winter | Medium-high | Weeded | 1.5932 | 0.1206 | A |
| | | Winter | Medium-low | Control | 1.56 | 0.1206 | A |
| | | Winter | Low | Control | 1.5307 | 0.1206 | A |
| | | Winter | High | Weeded | 1.5255 | 0.1206 | A |
| | | Spring | Medium-low | Weeded | 1.4318 | 0.0686 | A |
| | | Spring | Low | Weeded | 1.4287 | 0.0686 | AB |
| | | Spring | High | Control | 1.4042 | 0.0686 | ABC |
| | | Spring | Medium-high | Control | 1.3753 | 0.0686 | ABC |
| | | Spring | Medium-high | Weeded | 1.2951 | 0.06909 | ABC |
| | | Spring | Medium-low | Control | 1.2941 | 0.06909 | ABC |
| | | Spring | Low | Control | 1.2524 | 0.0686 | BC |
| | | Spring | High | Weeded | 1.2416 | 0.0686 | C |

**Table A3.** *Cont.*

| Effect | Dependent Variable | Season | Region | Estimate | SE | Letter | |
|---|---|---|---|---|---|---|---|
| Region | sqrt (x + 1) native species richness | Winter | Inland | 1.7882 | 0.1481 | A | |
| | | Winter | Coastal | 1.4283 | 0.1371 | A | |
| | | Spring | Inland | 1.5409 | 0.08366 | A | |
| | | Spring | Coastal | 1.1399 | 0.0774 | B | |

| Effect | Dependent Variable | Season | Region | Treatment | Estimate | SE | Letter |
|---|---|---|---|---|---|---|---|
| Region x Treatment | sqrt (x + 1) native species richness | Winter | Inland | Control | 1.8205 | 0.1526 | A |
| | | Winter | Inland | Weeded | 1.7559 | 0.1526 | A |
| | | Winter | Coastal | Weeded | 1.4566 | 0.1412 | A |
| | | Winter | Coastal | Control | 1.3999 | 0.1412 | A |
| | | Spring | Inland | Control | 1.5731 | 0.08635 | A |
| | | Spring | Inland | Weeded | 1.5087 | 0.08635 | A |
| | | Spring | Coastal | Weeded | 1.1899 | 0.07985 | B |
| | | Spring | Coastal | Control | 1.0899 | 0.07985 | B |

| Effect | Dependent Variable | Season | Treatment | Estimate | SE | Letter | |
|---|---|---|---|---|---|---|---|
| Treatment | sqrt (x + 1) non-native species richness | Winter | Control | 1.8419 | 0.05725 | A | |
| | | Winter | Weeded | 1.7391 | 0.05725 | B | |
| | | Spring | Control | 1.6285 | 0.06289 | A | |
| | | Spring | Weeded | 1.5494 | 0.06289 | A | |

| Effect | Dependent Variable | Season | Region | Estimate | SE | Letter | |
|---|---|---|---|---|---|---|---|
| Region | sqrt (x + 1) non-native species richness | Winter | Inland | 1.9098 | 0.07674 | A | |
| | | Winter | Coastal | 1.6712 | 0.07105 | B | |
| | | Spring | Inland | 1.669 | 0.08582 | A | |
| | | Spring | Coastal | 1.5089 | 0.07937 | A | |
| Region x initial native cover class | sqrt (x + 1) non-native species richness | Winter | Inland | Medium-low | 2.0247 | 0.09697 | A |
| | | Winter | Inland | Low | 1.8993 | 0.09697 | AB |
| | | Winter | Inland | Medium-high | 1.8841 | 0.09697 | AB |
| | | Winter | Inland | High | 1.831 | 0.09697 | AB |
| | | Winter | Coastal | Medium-low | 1.7201 | 0.08978 | AB |
| | | Winter | Coastal | High | 1.7143 | 0.08978 | AB |
| | | Winter | Coastal | Low | 1.6554 | 0.08978 | AB |
| | | Winter | Coastal | Medium-high | 1.5951 | 0.08978 | B |
| | | Spring | Inland | Medium-high | 1.7364 | 0.1045 | A |
| | | Spring | Inland | Medium-low | 1.7081 | 0.1045 | A |
| | | Spring | Inland | Low | 1.6992 | 0.104 | A |
| | | Spring | Coastal | Medium-low | 1.592 | 0.09624 | A |
| | | Spring | Coastal | Low | 1.592 | 0.09624 | A |
| | | Spring | Inland | High | 1.5323 | 0.104 | A |
| | | Spring | Coastal | High | 1.5005 | 0.09624 | A |
| | | Spring | Coastal | Medium-high | 1.3513 | 0.09624 | A |

## Appendix D  ANOVA Results for Established Shrub Metrics

**Table A4.** ANOVA results for established shrub metrics. Established shrub metrics include data on mean volume, total volume, and number of shrubs collected at the original ten restoration sites in 2012 and 2014. Treatment refers to whether plots experienced non-native plant removal (weeded) or not (control). Native cover class refers to the initial range of native cover for the plot (20–29% low, 30–39% medium-low, 40–49% medium-high, 50–59% high). Region refers to whether the restoration site was in the coastal or inland portion of Irvine Ranch Natural Landmarks. Mean shrub volume and shrub number data were ln(x)-transformed. Shaded rows highlight effects that were significant ($p < 0.05$).

| Response Variable | Effect | Num DF | Den DF | F Value | *p* Value |
|---|---|---|---|---|---|
| Mean shrub volume | Treatment | 1 | 135 | 3.28 | 0.0723 |
| | Native cover class | 3 | 135 | 1.2 | 0.3108 |
| | Native cover class × treatment | 3 | 135 | 1.06 | 0.3682 |
| | Region | 1 | 135 | 17.5 | <0.0001 |
| | Region × treatment | 1 | 135 | 1.49 | 0.2248 |
| | Region × native cover class | 3 | 135 | 0.07 | 0.9747 |
| | Region × native cover Class × treatment | 3 | 135 | 0.76 | 0.5166 |

**Table A4.** *Cont*.

| Response Variable | Effect | Num DF | Den DF | F Value | *p* Value |
|---|---|---|---|---|---|
| Total shrub volume | Treatment | 1 | 135 | 3.12 | 0.0796 |
| | Native cover class | 3 | 135 | 8.64 | <0.0001 |
| | Native cover class × treatment | 3 | 135 | 0.19 | 0.9031 |
| | Region | 1 | 135 | 0.73 | 0.3953 |
| | Region × treatment | 1 | 135 | 0.59 | 0.4455 |
| | Region × native Cover Class | 3 | 135 | 2.39 | 0.0718 |
| | Region × native Cover Class × treatment | 3 | 135 | 0.21 | 0.888 |
| Shrub number | Treatment | 1 | 135 | 9.95 | 0.002 |
| | Native cover class | 3 | 135 | 1.74 | 0.1626 |
| | Native cover class × treatment | 3 | 135 | 0.85 | 0.468 |
| | Region | 1 | 135 | 16.71 | <0.0001 |
| | Region × treatment | 1 | 135 | 1.01 | 0.316 |
| | Region × native cover class | 3 | 135 | 0.94 | 0.4244 |
| | Region × native cover Class × treatment | 3 | 135 | 1.76 | 0.157 |

## Appendix E Tukey Post Hoc Results for Established Shrub Metrics

**Table A5.** Tukey post hoc results for established shrub metrics. Results for (**a**) mean shrub volume, (**b**) total shrub volume, and (**c**) number of established shrubs on the effects of non-native plant removal treatment, native cover class, region, and all interactions. Treatment refers to whether plots experienced non-native plant removal (weeded) or not (control). Native cover class refers to the initial range of native cover for the plot (20–29% low, 30–39% medium-low, 40–49% medium-high, 50–59% high). Region refers to whether the restoration site was in the coastal or inland portion of Irvine Ranch Natural Landmarks. Mean shrub volume and shrub number data were ln(x)-transformed. Standard error (SE) and letters from the Tukey post hoc tests are included in results tables, with different letters indicating significant differences and shared letters indicating no significant difference amongst groups.

| (**a**) Tukey–Kramer post hoc results for ln(x) mean established shrub volume Effect of Region | | | |
|---|---|---|---|
| **Region** | **Estimate** | **SE** | **Letter** |
| Coastal | 5.5534 | 0.2199 | A |
| Inland | 4.253 | 0.2197 | B |

| (**b**) Tukey–Kramer post hoc results for total established shrub volume Effect of Initial Native Cover Class | | | |
|---|---|---|---|
| **Native Cover Class** | **Estimate** | **SE** | **Letter** |
| High | 24.6762 | 1.9698 | A |
| Medium-high | 21.6437 | 1.9698 | AB |
| Medium-low | 17.5053 | 1.9698 | B |
| Low | 17.4724 | 1.9767 | B |

| (**c**) Tukey–Kramer post hoc results for ln(x) number of established shrubs Effect of Treatment | | | |
|---|---|---|---|
| **Treatment** | **Estimate** | **SE** | **Letter** |
| Weeded | 2.7857 | 0.1437 | A |
| Control | 2.4356 | 0.1435 | B |

| Effect of Region | | | |
|---|---|---|---|
| **Region** | **Estimate** | **SE** | **Letter** |
| Inland | 3.152 | 0.1872 | A |
| Coastal | 2.0694 | 0.1873 | B |

## Appendix F  Logistic Regression Results for Established Shrub Mortality

**Table A6.** Logistic regression results for established shrub mortality. Shrub mortality was sampled at all thirteen restoration sites in 2014. Treatment refers to whether plots experienced non-native plant removal (weeded) or not (control). Native cover class refers to the initial range of native cover for the plot (20–29% low, 30–39% medium-low, 40–49% medium-high, 50–59% high). Region refers to whether the restoration site was in the coastal or inland portion of Irvine Ranch Natural Landmarks. Shaded rows highlight effects that were significant ($p < 0.05$).

| Response Variable | Effect | Estimate | SE | Z Value | *p* Value |
|---|---|---|---|---|---|
| Shrub mortality | (Intercept) | −1.22329 | 0.17106 | −7.1511 | $8.61 \times 10^{-13}$ |
| | Treatment | −0.35182 | 0.12274 | −2.8663 | 0.00415 |
| | Native cover class | −0.01446 | 0.5546 | −0.2607 | 0.79431 |
| | Region | −1.05432 | 0.14839 | −7.105 | $1.2 \times 10^{-12}$ |

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
