# Peer review of "Effects of Non-Native Annual Plant Removal on Native Species in Mediterranean-Climate Shrub Communities"

_diversity, doi:10.3390/d16020115_

Round 1
Reviewer 1 Report
Comments and Suggestions for Authors
This study encompasses data and observations beginning about 2010 and involving coastal and adjacent inland environments in an area of California that has long been heavily impacted by humans. This study identifies the importance of such a background, in addition to relevant literature bearing on the influence of non-native plant species density and diversity on native plant species composition. The study design was such that the authors were able to conclude that non-native plant species removal resulted in the increase in native shrub cover through increased native seedling recruitment, which is a potentially important finding. An emphasis on threshold levels of non-native plant species densities and management is also important.
This study convincingly found that non-native removal is most effective in environments where native cover is 20-39%. Where native cover is high, non-native removal has equivocal results for enhancing native cover. My concern here is that the way the paper is written (e.g., the needless use of the term "invasive" four times at the beginning of the paper, not really emphasizing their findings on threshold levels even in the Abstract where it is mentioned in the last sentence), managers will likely take the results of this study and generally make a universal conclusion that if non-natives have an impact in degraded environments, they are likely to have a negative impact in any environment including where native plant cover is high. I see over-management of non-native plants in high native cover environments as a big problem because it often leads to collateral damage of native plants (e.g., over-application or indiscriminate use of herbicide and removal programs).
One way the authors could address this is by mentioning that non-native plant species often have ecosystem services that should be balanced against disservices in any removal program. Also, at the bottom of page 12 and top of page 13 where reasons are given for not detecting much influence of non-native density on native forb density, it could also be that non-native plant species are often ecologically neutral to natives and simply filter into the environment and add to rather than replace standing plant diversity.
The paper is excellent and worthy of publication but it does have a tendency to take the easy route in simply condemning non-natives as invasive or something that should be always minimized, and this in spite of their excellent conclusions on thresholds.
Author Response
We have responded to the reviewers’ comments (details below, following each comment) and believe that the manuscript has been greatly improved as a result.
Reviewer 1
This study encompasses data and observations beginning about 2010 and involving coastal and adjacent inland environments in an area of California that has long been heavily impacted by humans. This study identifies the importance of such a background, in addition to relevant literature bearing on the influence of non-native plant species density and diversity on native plant species composition. The study design was such that the authors were able to conclude that non-native plant species removal resulted in the increase in native shrub cover through increased native seedling recruitment, which is a potentially important finding. An emphasis on threshold levels of non-native plant species densities and management is also important.
This study convincingly found that non-native removal is most effective in environments where native cover is 20-39%. Where native cover is high, non-native removal has equivocal results for enhancing native cover. My concern here is that the way the paper is written (e.g., the needless use of the term "invasive" four times at the beginning of the paper, not really emphasizing their findings on threshold levels even in the Abstract where it is mentioned in the last sentence), managers will likely take the results of this study and generally make a universal conclusion that if non-natives have an impact in degraded environments, they are likely to have a negative impact in any environment including where native plant cover is high. I see over-management of non-native plants in high native cover environments as a big problem because it often leads to collateral damage of native plants (e.g., over-application or indiscriminate use of herbicide and removal programs).
Response: This is an excellent point. We have removed the word “invasive” and replaced it with “non-native” and added some additional details regarding when the non-natives become invasive in this system (end of first paragraph of Intro and middle of second paragraph, beginning of third paragraph).
One way the authors could address this is by mentioning that non-native plant species often have ecosystem services that should be balanced against disservices in any removal program. Also, at the bottom of page 12 and top of page 13 where reasons are given for not detecting much influence of non-native density on native forb density, it could also be that non-native plant species are often ecologically neutral to natives and simply filter into the environment and add to rather than replace standing plant diversity.
Response: We have added a statement to this effect at the bottom of page 12 as recommended.
The paper is excellent and worthy of publication but it does have a tendency to take the easy route in simply condemning non-natives as invasive or something that should be always minimized, and this in spite of their excellent conclusions on thresholds.
Response: Thank you. We hope that our revisions address this concern.
Reviewer 2 Report
Comments and Suggestions for Authors
This paper investigates the effects of non-native plant removal on native species in Mediterranean-climate shrub communities. The findings have practical implications for land stewards and restoration efforts, highlighting the cost-effectiveness and labor intensity of non-native removal as a restoration method. However, there still are some suggestions to improve the manuscript.
Q1: Your scientific hypothesis serves as a proposed explanation for yourself scientific question, and I believe it should be based on previous research, summarized, and refined into a scientific theory, which is then tested and proven. Therefore, it is essential to provide additional scientific evidence for these hypotheses. It is advisable to refine your hypothesis to focus your research, usually by proposing only 1 or 2 hypotheses. This approach will help make your research more targeted and effective..
Q2: In the part of materials and methods, can we add a comparison picture of the type of native vegetation cover in the field, so as to make it more intuitive and clear?
Q 3: The salient marks between processing groups in the text boxes in FIG. 3b and FIG. 4 are unclear, are both coastal and inland salient or are only one of them salient?
Q 4: Could you further discuss the environmental conditions and factors that may influence the success of non-native removal, particularly in different regions?
Q 5: The removal of invasive plants is likely to lead to secondary invasion. It is important to consider whether other invasive species are present in the area or sample studied by the author. If so, the author should investigate the presence of other invasive species and, if not, consider how to avoid this secondary disaster. An in-depth discussion is warranted to address this issue comprehensively.
Q 6: At the end of the paper, it is suggested to add the conclusion part to summarize the research theme and put forward the prospect.
Q 7: Some volumes, periods, and pages of the references are incomplete, such as reference 77, please check and complete them all.
Comments on the Quality of English LanguageEnglish language fine
Author Response
Reviewer 2
This paper investigates the effects of non-native plant removal on native species in Mediterranean-climate shrub communities. The findings have practical implications for land stewards and restoration efforts, highlighting the cost-effectiveness and labor intensity of non-native removal as a restoration method. However, there still are some suggestions to improve the manuscript.
Q1: Your scientific hypothesis serves as a proposed explanation for yourself scientific question, and I believe it should be based on previous research, summarized, and refined into a scientific theory, which is then tested and proven. Therefore, it is essential to provide additional scientific evidence for these hypotheses. It is advisable to refine your hypothesis to focus your research, usually by proposing only 1 or 2 hypotheses. This approach will help make your research more targeted and effective..
Response: We have edited our research questions and hypotheses as recommended.
Q2: In the part of materials and methods, can we add a comparison picture of the type of native vegetation cover in the field, so as to make it more intuitive and clear?
Response: We have added these images as recommended.
Q 3: The salient marks between processing groups in the text boxes in FIG. 3b and FIG. 4 are unclear, are both coastal and inland salient or are only one of them salient?
Response: We added text to the legends of these figures to clarify that interactions were not listed because they were not significant. We did not run pair-wise comparisons.
Q 4: Could you further discuss the environmental conditions and factors that may influence the success of non-native removal, particularly in different regions?
Response: We edited the research questions to ask more generally about site conditions. We also added more information to the introductory paragraph regarding how site variation may influence the success of non-native removal.
Q 5: The removal of invasive plants is likely to lead to secondary invasion. It is important to consider whether other invasive species are present in the area or sample studied by the author. If so, the author should investigate the presence of other invasive species and, if not, consider how to avoid this secondary disaster. An in-depth discussion is warranted to address this issue comprehensively.
Response: We agree that this is an important concern. It is mentioned in the Discussion section (thresholds paragraph). We also added a description of this challenge to the 4th paragraph of the introduction. An in-depth discussion is outside the scope of this paper, as we were working with sites that had at least 20% native cover.
Q 6: At the end of the paper, it is suggested to add the conclusion part to summarize the research theme and put forward the prospect.
Response: We have added a conclusion statement as recommended.
Q 7: Some volumes, periods, and pages of the references are incomplete, such as reference 77, please check and complete them all.
Response: Thank you for pointing this out. We have gone through the references and corrected errors.
Reviewer 3 Report
Comments and Suggestions for Authors
This Ms is well organized and well written.
The authors should clarify the non-native plants and non-native invasive plants in both the title and the text. Non-native plants and non-native invasive plants are different. But in this Ms, they are mixed. Sometimes For example, non-native plants (Line 67), sometimes non-native invasive plants (Line 52, Line 76). Particularly, you should clarify the species you removed are invasive or non-invasive alien plants (Line 177-178). As I know some of them are severe invasive alien plants in California.
In the M & M. What did you remove from the weeded plots. Please provide details. Besides the growing non-native plants, how about the litter or thatch of the invasive species? Did you remove the thatch in the plots? As I know, the accumulating thatch in the invaded regions play an important role (Chen B-M, D’Antonio CM, Molinari N, Peng S-L (2018) Mechanisms of influence of invasive grass litter on germination and growth of coexisting species in California. Biological Invasions 20:1881-1897; Molinari NA, D'Antonio CM (2014) Structural, compositional and trait differences between native- and non-native-dominated grassland patches. Functional Ecology 28:745-754; Molinari NA, D’Antonio CM (2020) Where have all the wildflowers gone? The role of exotic grass thatch. Biological Invasions 22:957-968.).
The authors should list the references in uniform format. Some information (volum or pages) is missed (e.g. the last ref.).
Author Response
Reviewer 3
This Ms is well organized and well written.
The authors should clarify the non-native plants and non-native invasive plants in both the title and the text. Non-native plants and non-native invasive plants are different. But in this Ms, they are mixed. Sometimes For example, non-native plants (Line 67), sometimes non-native invasive plants (Line 52, Line 76). Particularly, you should clarify the species you removed are invasive or non-invasive alien plants (Line 177-178). As I know some of them are severe invasive alien plants in California.
Response: Reviewer 1 also made this excellent point. We have removed the word “invasive” and replaced it with “non-native” and added some additional details regarding when the non-natives become invasive in this system (end of first paragraph of Intro and middle of second paragraph, beginning of third paragraph).
In the M & M. What did you remove from the weeded plots. Please provide details. Besides the growing non-native plants, how about the litter or thatch of the invasive species? Did you remove the thatch in the plots? As I know, the accumulating thatch in the invaded regions play an important role (Chen B-M, D’Antonio CM, Molinari N, Peng S-L (2018) Mechanisms of influence of invasive grass litter on germination and growth of coexisting species in California. Biological Invasions 20:1881-1897; Molinari NA, D'Antonio CM (2014) Structural, compositional and trait differences between native- and non-native-dominated grassland patches. Functional Ecology 28:745-754; Molinari NA, D’Antonio CM (2020) Where have all the wildflowers gone? The role of exotic grass thatch. Biological Invasions 22:957-968.).
Response: We clarified in the Methods section that thatch was not intentionally removed along with non-native removal, but that it resulted in a reduction in thatch and an increase in bare ground. We added citations to the studies mentioned in the Introduction.
The authors should list the references in uniform format. Some information (volum or pages) is missed (e.g. the last ref.).
Response: Thank you for pointing this out. We have gone through the references and corrected errors.
Round 2
Reviewer 2 Report
Comments and Suggestions for Authors
The author's revision has significantly improved the article, and I have no further concerns.